# Interaction of Corporate Social Responsibility Reporting at the Crossroads of Green Innovation Performance and Firm Performance: The Moderating Role of the Enterprise Life Stage

**Fawad Rauf** [1,2], **Wanqiu Wang** [1,*] **and Cosmina L. Voinea** [3,*]

1 College of Economics and Management, Beijing University of Technology, Beijing 100124, China; fawadraufkhan@gmail.com
2 School of Management, Xi'an Jiaotong University, Xi'an 710049, China
3 Faculty of Management, Open University of the Netherlands, P.O. Box 2960, 6401 DL Heerlen, The Netherlands
* Correspondence: wangwanqiu@bjut.edu.cn (W.W.); cosmina.voinea@ou.nl (C.L.V.)

**Abstract:** This research delves into the intricate interplay between green innovation performance (GIP), firm performance (FP), and corporate social responsibility (CSR) reporting, leveraging enterprise life stage performance as a pivotal moderator. Analyzing a robust sample of 5450 firm-year observations spanning from 2015 to 2021, this study employs OLS regressions with panel data sourced from the CSMAR and HEXUN databases to validate prevailing research hypotheses. The findings underscore the pivotal role of CSR reporting in augmenting corporate value while concurrently mitigating inadequacies within the system. Moreover, this study uncovers a nuanced relationship between CSR reporting, GIP, and FP in the context of China, revealing a significant moderation effect attributed to the enterprise life cycle. These revelations carry profound implications for CSR reporting stakeholders, including academics, practitioners, and regulators. Notably, they provide valuable insights to authorities and boards of directors concerning the growth potential of enterprises and states. A distinctive facet of this study lies in its exploration of the moderating influence of an enterprise's life stage on the relationship between CSR reporting and GIP or FP.

**Keywords:** firm performance; green innovation performance; CSR reporting; enterprise life stage

## 1. Introduction

The expanding worldwide concerns for environmental sustainability, as well as the need for new solutions, has prompted considerable interest across sectors, academia, local government, and numerous institutions [1,2]. The recent corporate landscape, typified by increased investor scrutiny and expectations, has presented a new set of problems to firms and financial markets. Corporate social governance (CSR) concerns have swiftly progressed from being mere considerations to being basic pillars of corporate decision making, establishing themselves as critical determinants for the public good [3]. CSR reporting and performance have emerged as key considerations in the twenty-first century. They have piqued the interest of a wide range of stakeholders, from investors and financial institutions to governments all around the world. The emphasis on CSR reporting stems from an understanding that a company's success in these areas not only adds to its ethical standing but also has important consequences for its financial viability and long-term sustainability. CSR reporting safeguards shareholder rights, decision-making authority, and organizational governance. Much research on CSR and firm performance (FP) has been available since the 1960s [4]. According to Michelon et al. [5], both CSR practices and CSR reporting are advantages for businesses. This type of CSR reporting becomes necessary and useful for businesses to analyze the impact of CSR reporting on their image and acquire a competitive edge [5].

Two key aspects are important to this dynamic: company performance and green innovation [6]. As considerable earlier research has revealed, these factors have a significant impact on CSR. Firm success, as measured by quantifiable measures of financial health, is critical in influencing a company's CSR reporting. Extensive research has shed light on the link between business performance and CSR reporting, emphasizing the importance of strong financial indicators in improving CSR disclosures [7]. Green innovation, which exemplifies sustainable and environmentally responsible corporate practices, has emerged as a viable path for tackling the issues provided by CSR standards [8].

Green innovation performance (GIP) has received increasing attention in recent decades due to the benefits it provides for resource conservation, environmental preservation, and firm performance. GIP is a company's most critical and proactive approach to environmental development. Green innovation performance (GIP) indicates market shifts. Buisson and Silberzahn [9] define GIP as technical improvements that conserve energy and reduce pollution. As a result, fast economic growth and advancement are crucial to resolving concerns of corporate social governance. Pollution, water scarcity, and other environmental issues have lately emerged as worldwide concerns [9,10].

However, the synergy between green innovation and firm performance remains a complex riddle, with much discussion within the academic community. This difficult position of green innovation in terms of synergy with financial success presents important challenges, particularly when seen through the perspective of an enterprise life stage [11,12]. A company's enterprise cycle is a critical contextual aspect that strongly determines its objectives, capabilities, and strategic decisions regarding CSR reporting. Companies at various points of their life cycle stage have different problems and make different decisions, which have differing ramifications for their approach to CSR reporting. Despite its importance, this aspect has received little attention in the available literature [13,14].

Prior research had both theoretical and practical deficiencies. Many studies have relied on subjective measurement methods, have lacked empirical evidence to elucidate the intricate interplay between green innovation, firm performance, and CSR reporting, and have inadequately considered the influence of the enterprise life stage [15–17]. This study seeks to fill these gaps by investigating the influence of financial and green innovation performance on a company's CSR reporting. It also aims to determine whether this effect varies across different periods of an enterprise's life stage. To perform this, this study used a thorough examination of green patent data, financial data, and CSR data from Chinese A-share-listed businesses during a certain time period.

The key goals of this research are twofold: to explore the impact of financial and green innovation success on a company's CSR reporting and to establish whether this impact varies depending on the stage of the organization's life cycle. This research aims to contribute to a more nuanced understanding of the complex interplay between financial and green innovation performance and CSR reporting, particularly as influenced by contextual factors. This research intends to provide helpful direction to both theory and practice by giving empirical insights and supporting organizations and investors in navigating the growing environment of sustainability and responsible business practices.

We expanded on a prior study [18] based on the perspectives of life cycle theories, stakeholder theories, and legitimacy theories, while exploring the impact of business performance on CSR. CSR reporting and FP have been linked in prior research positively or negatively, indicating that other variables may affect or cause their relationship [19]. This paper builds on prior research by examining the enterprise life stage (ELS), stakeholders, and legitimacy theories [20]. These findings suggest that other factors might be contributing to their association [20]. Research has analyzed the ELS from various angles [21,22]. A previous study has overlooked the ability of CSR reporting to influence business performance using the ELS as a moderating factor. In this study, the ELS is used as a moderator to examine the impact of CSR reporting on business performance. As a result, we explain how the company's ELS is a mediator between CSR reporting and FP.

The contributions of this work are dual, encompassing theoretical and practical aspects, producing useful implications for academia and the commercial sector. At the theoretical level, this research creates new ground by digging into the delicate link between firm performance, green innovation, and corporate social governance (CSR) reporting. It offers a new contextual component, the enterprise life stage, which had hitherto gone unnoticed in literature. By experimentally proving the moderating impact of an enterprise's life stage on the effects of financial and green innovation success on CSR reporting, this study extends the theoretical framework in the field of corporate sustainability. Additionally, the research bridges the gap between theory and reality by applying a strong empirical technique using green patent data from Chinese A-share-listed businesses over an extended time. This method not only provides factual facts but also offers a holistic picture of how finance and innovation strategies are intertwined with CSR achievement. This study's practical consequences are similarly substantial. Companies stand to gain from greater knowledge of how to strategically link their financial and green innovation objectives with their CSR reporting, taking into consideration their individual life cycle stages. This understanding helps firms to make educated decisions about the problems and opportunities that exist at various stages of their development. For investors, the results give a helpful tool to measure a company's sustainability performance, permitting better-educated investment decisions that consider a firm's financial stability, dedication to green innovation, and alignment with CSR ideals. This study bridges the gap between theory and practice, providing both businesses and investors with the knowledge needed to navigate the ever-changing environment of responsible and sustainable corporate practices. Finally, the research not only increases academic knowledge but also provides a practical path for firms and investors looking to flourish in an era where sustainability and ethical business practices are critical.

This paper is organized as follows. Section 2 presents the theoretical framework and hypothesis development. Section 3 shows an overview of the research design, which includes a sample description, definitions of variables, and analyses utilized. Section 4 shows the key empirical findings. Finally, there are some closing observations in Section 5.

## 2. Theoretical Framework and Hypothesis Development

In the research, factors relating to CSR reporting are considered explanatory variables that help with predicting business performance. We are also looking into how CSR information, such as financial and green innovation, performs under the contextual role of ELC.

### 2.1. Relationship between CSR Reporting and Firm Performance

CSR has evolved into a critical concern for stakeholders and investors, spanning social, environmental, and economic dimensions [23]. Numerous research has shown that corporate performance has a major influence on CSR ratings [24]. CSR reporting may be used to differentiate across otherwise equivalent enterprises or nations [25]. As a result, practitioners frequently incorporate CSR reporting data into their investing strategies, potentially resulting in modifications in security weightings [26]. Furthermore, legitimacy theory illuminates the relationship between environmental reputation and the financial implications of socially responsible behavior [27,28]. This approach considers firms to be important parts of a larger societal system [27]. However, the integration of CSR reporting into firm performance and corporate strategy is presently gaining traction, owing in part to the difficulties shareholders confront in acquiring CSR reporting data at the firm level.

The relevance of company performance on CSR rating factors has received a lot of attention. Early research indicated that investments in environmentally or socially responsible projects might have an influence on corporate performance [29]. Empirical studies have repeatedly shown a correlation between a company's CSR disclosures and its business performance or valuation [30]. Furthermore, some studies have found that CSR ratings are related to non-financial performance indicators, such as enhanced process efficiency, reduced material consumption, and energy conservation [31].

To guide investment decisions, CSR rating information frequently depends on qualitative CSR analysis. Proprietary CSR ratings are created using both internal and third-party research, providing a thorough overview of CSR information for each firm in the portfolio and investable universe [32]. Some CSR rating practitioners, however, question these private ratings and investigate the potential effects of CSR rating data on its reputational worth. This dynamic has fueled a desire for competitive advantage through innovation, which has been strengthened by green legitimacy [33]. However, El Ghoul and Karoui (2020) discovered that FP had a favorable impact on CSR ratings in 53 countries [34,35]. Despite the ever-changing landscape of the environmental literature, there is widespread consensus that FP has a favorable influence on CSR rating in terms of overall company performance [36]. As a result, our initial hypothesis is.

**Hypothesis H1:** *There is a positive relationship between CSR reporting and firm performance.*

*2.2. Link between CSR Reporting and Degree of Green Innovation Performance*

The growing interest in green innovation performance (GIP) has piqued the curiosity of businesses operating in today's intensely competitive economy, where capturing significant market share and a competitive edge is critical. GIP's appeal stems from its promise to aid environmental preservation and resource conservation. Companies are typically viewed as the principal contributors to environmental concerns, and they face significant pressure from a varied range of investors in terms of environmental legitimacy [37]. GIP aims to reduce pollution, increase energy production, reduce waste creation, substitute sustainable alternatives for scarce resources, and promote recycling [38]. Businesses are actively investigating techniques to minimize material and energy consumption across their manufacturing processes, adopt recycling procedures for old materials, and reduce waste and material disposal after production. GIP administration has grown in importance in both business and academics, notably in the context of CSR reporting. As a result, academics' attention has switched from just finding the components that drive GIP to investigating its influence on a company's social performance [39].

Furthermore, a thorough analysis of the earlier GIP literature, including important researchers, institutions, and significant publications, contributes to understanding the multidimensional idea of green innovation performance. This study highlights three main aspects of GIP. Sustainable innovation includes both social and environmental components and is typically associated with CSR reporting as a comparable indicator [40]. Companies seeking significant market shares and competitive advantages are especially drawn to GIP, which is typically dependent on their CSR rating and local context. Businesses that prioritize creative product designs targeted at decreasing energy consumption during consumer usage, reducing post-consumption waste or eliminating dangerous components, are seen to perform better in terms of CSR. Environmental reporting correlates with GIP [41,42].

Policymakers and academics underline the importance of GIP in effectively tackling environmental concerns and promoting company sustainability [43]. As a result, businesses are becoming more conscious of the environmental effects of their decision making and management practices, aggressively pushing GIP [41]. As a result, we propose a hypothesis.

**Hypothesis H2:** *There is a positive relationship between CSR reporting and green innovation performance.*

*2.3. CSR Reporting, Firm Performance, and Green Innovation Performance: The Enterprise Life Stages as a Moderator*

The enterprise life stage (ELS) idea compares a business to a living organism that goes through several stages such as birth, development, maturity, and decline [41,44,45]. Prior research has proven that organizations function in diverse settings and financial situations at different periods of their life stage, demanding appropriate business strategies [15,46]. This idea applies to CSR practices and transparency initiatives. Corporations must dynamically

connect their CSR policies with financial outcomes based on the stage of their company's life cycle. In this study, ELS stages were used as a moderator for the first time. The ELS appears to be an important component in determining CSR ratings and company performance. The ELS emerges as a critical variable influencing CSR rating and firm performance [17]. The maturity of a corporation has a considerable impact on the allocation of time and resources to CSR, which is determined by a variety of factors across different ELS. One important feature of this is that older organizations have more stable and predictable capital flows than their younger counterparts [15]. This notion encourages increasing expenditures for older organizations, even if they may not see CSR spending as a requirement owing to their established stability. According to the findings, younger enterprises may be more suited for CSR initiatives [47]. However, growing enterprises may have cash flow restrictions, such as low liquidity ratios, limiting their capacity to compete in CSR activities with established firms [48,49]. Profiting while raising capital and preparing for future cash flows becomes difficult when enterprises must obtain costly finance [50]. Enterprises emphasize survival in the early phases of their ELS and are less concerned with topics like CSR, ESG, non-direct expenditures, goodwill shocks, and financial reporting consequences. Instead, they concentrate on obtaining the capital required for subsistence. This includes cost-effective and creative investments, steady cash flows, and growth scenarios that attract early-stage businesses [15]. As a company grows, it may increase its investments and improve its image using resources and competencies that younger enterprises do not have [15]. When compared to young enterprises, maturity is related to improved financial ratios, larger dividends, and increased cash inflows [51]. Furthermore, mature firms are more conscious of critical challenges, such as their reputation and contacts with important stakeholders and authorities. Because of their vested interests, mature organizations are more involved in CSR initiatives [52]. As a result, the following hypotheses are presented:

**Hypothesis (H3a):** *Enterprise life stages have a negative effect on CSR reporting.*

**Hypothesis (H3b):** *Enterprise life stages have a negative moderating effect on CSR reporting and firm performance.*

**Hypothesis (H3c):** *Enterprise life stages moderate the link between GIP and CSR reporting.*

### 3. Data, Measurement, and Research Methodology

*3.1. Sample*

China was chosen as this study's case study. In 2009, 335 companies were required to report on CSR. The Shanghai Stock Exchange (SZSE) and Shenzhen Stock Exchange (SHSE) yearly financial reports and CSR reports provide CSR-related corporate data. The China Stock Market & Accounting Research (CSMAR) databanks and the HEXUN database provide financial data, corporate governance, and GIP data. Unmatched and matched sample analyses were both performed by us. We acquired an odd sample of 5450 firm-year observations with appropriate data during the 2015–2021 period that show 2022 separate companies after combining the above datasets and deleting observations missing needed characteristics. The treatment sample has 2655 firm-year observations.

*3.2. Dependent Variable and Independent Variable*

3.2.1. Corporate Social Governance (CSR) Reporting

We utilized the sustainability ratings of Chinese-listed firms from the HEXUN website according to Shahab et al. [53]. We were able to obtain the RKS ratings for Chinese company corporate social governance factors using this website. All Chinese businesses that publish sustainability reports and receive annual ratings from reputable organizations are included in the HEXUN database. Using the HEXUN-RKS ratings for each sustainability component, we evaluated the rating of the performance in the areas of governance, social, and environmental sustainability.

### 3.2.2. Firm Performance (FP)

We used two optional performance measures as the independent variable because there is no single standard measure for firm performance (FP). Different performance characteristics were utilized to assess firm values, and a few criteria, such as investors' equity, assets, and net profits/total profits after these costs, were used to measure FP from various angles. We used ROA (return on assets) and ROE (return on equity) as firm performance indicators in this study. ROE considers operating income split by shareholders' equity and earnings before interest and tax (EBIT) over resources [54–56]. ROA measures shareholder value by the efficiency of a company's assets, whereas ROE is a regularly used metric in the corporate governance literature that evaluates the company's operational performance.

### 3.2.3. Green Innovation Performance (GIP)

This study takes GIP as an independent variable gauged on a pair of components: intention and performance. The China National Holding Administration (CNIPA) has combined 10,137 application patents from 327 listed energy-intensive firms, including patents, efficacy models, etc. Furthermore, 2971 green patents were removed to obtain the following keywords: (1) environmentally friendly, (2) low carbon, (3) ecology, (4) low carbon, (5) ecology, (6) energy efficient, (7) cost effective, (8) keep it clean, and (9) recycle. The 10,137 evaluation patents resulted in (10) environmental protection and (11) emissions reduction. This study measures GIP by quantifying a company's green patents as two variables that equals one for companies with at least one green patent and zero otherwise [57].

### 3.3. Moderating Variable: Enterprise Life Stages (ELS)

An important moderating variable used for this study was the ELS. The academic literature offers various framework for ccategorizing the stage of the ELS leading lack of consensus on the optimal approach for delineating this period [15]. Recognizing the diversity of perspective on the phases of ELS, our study dose not commit to a predefined number of stages. We adopt the measure of the retained earnings to total assets (RETA) following the approach by DeAngelo et al. [58] as a proxy for identifying the firm stage with enterprise life stages. The RETA ratio serves as an indicator of a firm's reliance on internal versus external financing, with higher RETA value suggesting a mature stage characterize by reduced investment opportunity. And lower RETA values indicating a firm in its younger, growth-oriented stages [59].

### 3.3.1. Control Variables

When examining the relationship between social and moral activities and business values, several factors are to be considered. Some researchers have used one or more control variables to get rid of or reduce the influence of the DV on the IV. Thus, according to the previous study, we incorporated various control factors into our model related to the organization and its surroundings.

In this study, we addressed various factors that could potentially influence CSR reporting, as identified by previous research [7,60,61]. These factors include firm age (FA). Firm age is defined as the number of years since the establishment of the company, as established by Guo et al. [62]. Financial leverage (FL): Financial leverage is considered a potential predictor of financial performance (FP) and is instrumental in resolving agency problems within public entities. Prior studies have typically calculated financial leverage using the debt-to-equity ratio, as shown by Yeh et al. [63]. State-owned enterprises (SOEs): State-owned enterprises were assessed using a specific factor, with a value of one denoting firm regulated or owned by the state or government, following the approach of Zhu et al. [64]. Chief executive officer duality (CEO duality): CEO duality pertains to situations where the CEO also serves as the company's board chair, potentially affecting corporate governance and the concentration of power within the CEO, as observed in a study by Zhou [65]. This variable is represented as a dummy variable, with one indicating CEO duality and zero indicating its absence. Firm size (FS): Firm size is employed as an

indicator of financial performance and credibility. We assessed firm size using metrics such as net income (asset) and employee count, as previously utilized in studies like de Abreu et al. [66]. Ownership concentration (OC): Ownership concentration was calculated based on the total stockholding of a primary creditor, as demonstrated in research by Rehman et al. [67]. Investment opportunities (IOs): Investment opportunities are defined as a firm's market value multiplied by the replacement value of its assets, as per the definition by Zhao [65]. Growth opportunity (GO): Growth opportunity is described as the rate of increase in the company's primary income, as outlined in a study by Li and Zhang [68]. Industry (industry dummy): Industry dummies were incorporated to control the industry-specific effects on the company. Year (year dummy): Year dummies were retained to account for potential unobserved threats that may impact a company's performance over time. These factors have been widely used in studies involving Chinese companies to enhance the robustness and comprehensiveness of our analysis.

### 3.3.2. Estimating Model

We apply Baron and Kenny's approach to analyze the mediating role of GIP on the link between CSR reporting and FP. Kenny et al. [69] found that a mediator is a component that helps to explain the link between independent and outcome variables. Following Baron and Kenny [69], a variable in the first equation must influence a variable in the second equation. For the second equation to demonstrate mediation, the IV must influence the mediator variable; for the third equation to explain mediation, the mediator must affect the DV. If all these requirements hold in the predicted direction, the influence of the IV on the DV in the third equation must be smaller than in the second [69–71]. In other words, as indicated by Rauf et al. [28]), there are four phases used in producing mediation.

We built ordinary list square (OLS) regression models to test our hypothesis and then used fixed effect tests to investigate further. We used struggling explanatory variables to reduce endogenous problems to develop empirical models.

Based on Model 1, the following relationship is examined between FP and CSR reporting:

$$CSR\ Reporting = a + \beta_1 FP + \sum_{i=1}^{N} \beta_n controls_{(i,t)} + \varepsilon_{(i,t)} \tag{1}$$

Model 2 is used to examine the impact of CSR reporting on corporate GIP:

$$CSR\ Reporting_{(i,t)} = a + \beta_2 GIP + \sum_{i=1}^{N} \beta_n controls_{(i,t)} + \varepsilon_{(i,t)} \tag{2}$$

Model 3 is used to examine the impact of the ELS moderating the relationship between FP and CSR reporting.

$$CSR\ Reporting_{(i,t)} = a + \beta_3 FP + \beta_5\ FP \times ELS + \sum_{i=1}^{N} \beta_n controls_{(i,t)} + \varepsilon_{(i,t)} \tag{3}$$

Model 4 is used to examine the impact of the ELS moderating the relationship between GIP and CSR reporting.

$$CSR\ Reporting_{(i,t)} = a + \beta_6 GIP + \beta_7 GIP \times ELS + \sum_{i=1}^{N} \beta_n controls_{(i,t)} + \varepsilon_{(i,t)} \tag{4}$$

where (CSR) reporting indicates a company's corporate social responsibility reporting; (GIP) refers to green innovation performance; (FP × ELS) shows the interaction between FP and enterprise life stages; (GIP × ELS) shows the interaction between GIP and ELS, where i and t denote firm and year, respectively; β stands for the presumed parameter; and $\sum$ indicates the error term, while the controls refer to firm-level control variables.

## 4. Empirical Results

### 4.1. Descriptive Statistic

Table 1 summarizes the statistical properties of the independent, dependent, moderating, and controlling factors. CSR rating averages 5.625, with a standard deviation of 1.567, indicating an intermediate level between BB and AA. Firm performance, as measured by FP (ROA), is 0.038 on average, with a standard deviation of 0.048. The green innovation performance (GIP) score has an average of 1.345, with a standard deviation of 22.237. The mean enterprise life stage (ELS) value is 20.040, with a standard deviation of 0.895. Other variables with average values of 2.156, 0.447, 0.476, 0.227, 20.056, 0.356, 0.043, and 0.268 are firm age (FA), financial leverage (FL), state-owned enterprise (SOE) dummy, chief executive officer duality (CEOD) dummy, firm size (FS), ownership concentration (OC), investment opportunity (IO), and growth opportunity (GO), respectively.

**Table 1.** Descriptive statistics.

| Variables | Mean | SD | Min | Max |
|-----------|------|------|------|------|
| CSR Reporting | 5.625 | 1.567 | 1.000 | 1.000 |
| FP (ROA) | 0.038 | 0.048 | −0.185 | 0.175 |
| FP (ROE) | 0.047 | 0.085 | −0.184 | 0.170 |
| GIP | 1.345 | 20.040 | 0.000 | 59.000 |
| ELS | 2.027 | 0.895 | 0.000 | 1.000 |
| FA | 2.156 | 0.758 | 0.000 | 3.121 |
| FL | 0.447 | 0.278 | 0.076 | 0.980 |
| SOEs | 0.476 | 0.493 | 0.000 | 1.000 |
| CEOD | 0.227 | 0.412 | 0.000 | 1.000 |
| FS | 20.056 | 1.268 | 19.546 | 25.840 |
| OC | 0.356 | 0.183 | 0.090 | 0.740 |
| IO | 0.043 | 0.080 | 0.180 | 0.230 |
| GO | 0.268 | 0.594 | −0.454 | 4.045 |

### 4.2. Correlation Matrix

The correlations between independent variables in our regression model are shown in Table 2. In general, Pearson correlations between independent variables are modest. For example, CSR reporting information releases and the score of responsible governance have a high connection of 0.456. The correlation coefficient between the degree of GIP and the score for accountable governance has the most significant value (0.264). The variance inflation factor (VIF) is minimal, indicating that our model's independent variables are not multicollinear. In Table 3, we show the distributional statistics and the univariate differences of variables of the samples from China. There is a considerable difference between the two samples in positions of CSR reporting and the degree of GIP, FP, and ELS.

**Table 2.** Correlations matrix.

| Variables | (1) | (2) | (3) | (4) | (5) | (6) | (7) | (8) | (9) | (10) | (11) | (12) | (13) |
|---|---|---|---|---|---|---|---|---|---|---|---|---|---|
| CSR Reporting | 1.000 | | | | | | | | | | | | |
| FP (ROA) | 0.090 *** | 1.000 | | | | | | | | | | | |
| FP (ROE) | 0.310 ** | 0.407 *** | 1.000 | | | | | | | | | | |
| GIP | 0.246 ** | 0.213 *** | 0.225 *** | 1.000 | | | | | | | | | |
| ELS | −0.134 * | 0.030 * | 0.026 * | 0.012 * | 1.000 | | | | | | | | |
| FA | 0.073 ** | 0.073 *** | 0.274 *** | 0.086 *** | 0.072 * | 1.000 | | | | | | | |
| FL | 0.054 *** | 0.134 *** | 0.387 *** | 0.143 *** | 0.324 *** | 0.042 *** | 1.000 | | | | | | |
| SOEs | 0.164 *** | 0.060 *** | 0.364 *** | 0.109 *** | 0.293 *** | 0.106 *** | 0.104 *** | 1.000 | | | | | |
| CEOD | 0.075 *** | 0.025 ** | 0.168 *** | 0.031 *** | 0.136 *** | 0.214 *** | 0.159 *** | 10.124 ** | 1.000 | | | | |
| FS | 0.256 *** | 0.080 ** | −0.074 ** | 0.092 ** | 0.381 *** | 0.438 *** | 0.134 ** | 0.365 *** | 0.216 *** | 1.000 | | | |
| OC | −0.236 ** | −0.175 ** | 0.167 ** | 0.097 ** | −0.456 ** | 0.199 ** | −0.037 ** | −0.024 * | 0.094 ** | 0.091 ** | 1.000 | | |
| IO | −0.023 ** | 0.063 ** | −0.129 ** | −0.038 ** | 0.072 ** | 0.056 ** | 0.012 * | −0.089 ** | −0.016 | −0.064 ** | 0.070 *** | 1.000 | |
| GO | 0.367 ** | 0.335 ** | −0.294 ** | 0.025 ** | 0.015 * | −0.018 * | −0.009 | 0.028 ** | 0.153 ** | −0.039 ** | 0.084 ** | 0.064 ** | 1.000 |

*, **, ***, significant at 10%, 5%, and 1%, respectively.

### 4.3. Regression Results

The OLS regression results in Equations (1)–(3) are presented below in Table 3. First, in Model 1, the independent variable FP (ROA) must be significantly related to the dependent variables in Model 1, which showed that FP was significantly and positively associated with firm CSR reporting ($\beta$ = 4.062, $p < 0.000$), confirming Hypothesis (H1), which is in line with the study of [72,73].

Secondly, the second model relates the GIP variable to CSR reporting. Table 3 shows that GIP positively impacts CSR reporting ($\beta$ = 0.0472, $p < 0.001$). When GIP was added, the company's CSR reporting was still positively affected, which is in line with the literature [74,75].

Table 3 presents the results of estimating Model 3 to test our Hypothesis (H3a). To define the role of the "degree of the ELS", the regression of CSR reporting as a dependent variable is depicted in Table 3. The results of the OLS analysis are reported in Table 3. Our findings highlight a negative and significant relationship between ELS and CSR reporting, confirming the research Hypothesis (H3a). Table 3 presents Model 4, the estimation results of the moderating effect of the ELS on the association between CSR reporting and GIP. The ELS is negatively correlated to CSR reporting ($\beta$ = −0.485, $p < 0.000$). The estimation coefficients of the interaction term ELS care are all significant but with opposite signs compared with those of CSR reporting, indicating that the ELS connection assuages the effect of CSR reporting. Thus, Hypothesis (H3b) is supported and is in line with [13,15].

**Table 3.** Moderating effect of enterprises life stage on the relation between green innovation performance, firm performance, and CSR reporting.

| Variables CSR Reporting | Model 1 FP (ROA) | Model 2 GIP | Model 3 ELS | Model 4 FP × ELS | Model 5 GIP × ELS |
|---|---|---|---|---|---|
| FP (ROA) | 4.062 ***<br>(0.326) | — | — | 2.671 ***<br>(6.876) | — |
| GIP | — | 0.0472 ***<br>(0.038) | — | | 0.0491 ***<br>(0.047) |
| ELS | — | — | −0.0532 ***<br>(0.045) | −0.0483 ***<br>(0.039) | −1.107 ***<br>(8.683) |
| FP × ELS | — | — | — | −0.485 ***<br>(0.040) | — |
| GIP × ELS | — | — | — | — | −1.109 ***<br>(8.664) |
| FA | 0.012 ***<br>(2.916) | 0.017 ***<br>(2.925) | 0.013 ***<br>(2.923) | 0.014 ***<br>(2.952) | 0.018 ***<br>(2.927) |
| FL | −1.105 ***<br>(−8.662) | −0.832 ***<br>(−6.169) | −1.105 ***<br>(−8.662) | −0.849 ***<br>(6.172) | −0.834 ***<br>(6.167) |
| SOEs | 0.971 **<br>(2.243) | 0.917 **<br>(2.396) | 0.918 **<br>(2.397) | 0.939 **<br>(2.472) | 0.918 **<br>(2.398) |
| CEOD | −0.352<br>(−0.875) | −0.402<br>(−0.993) | −0.406<br>(−0.994) | −0.407<br>(−0.987) | −0.403<br>(−0.978) |
| FS | 1.025 ***<br>(41.672) | 1.015 ***<br>(40.923 | 1.012 ***<br>(40.921) | 1.014 ***<br>(40.952) | 1.016 ***<br>(40.928) |
| OC | 0.018 ***<br>(3.102) | 0.021 ***<br>(0.243) | 0.025 ***<br>(0.247) | 0.027 ***<br>(0.258) | 0.025 ***<br>(0.247) |
| IO | −0.010 ***<br>(3.713) | −0.011 ***<br>(3.951) | −0.011 ***<br>(3.951) | −0.013 ***<br>(3.974) | −0.015 ***<br>(3.958) |
| GO | 0.004 ***<br>(7.111) | 0.005 ***<br>(7.110) | 0.005 ***<br>(7.110) | 0.006 ***<br>(7.118) | 0.007 ***<br>(7.115) |
| YI | included | included | included | Included | included |
| Constant | 10.355 ***<br>(30.591) | 10.838 ***<br>(8.662) | −1.105 ***<br>(−8.662) | −7.873 ***<br>(−20.283) | −9.838 ***<br>(19.256) |
| R² | 0.654 | 0.657 | 0.659 | 0.683 | 0.695 |

Note: **, ***, significant at 5%, and 1%, respectively.

Finally, in Model 5, we estimate Hypothesis (H3c) to see how it relates to the ELS. The results confirm a negative relationship between the moderating role of the ELS and

CSR reporting. Model 5's results show the coefficient for interaction when its GIP × ELS interaction term is negative and significant with FP ($\beta = -1.109$, $p < 0.000$), confirming Hypothesis (H3c), which is in line with the literature [76].

We have concluded that GIP regulates the association between CSR reporting and FP to a certain extent, which supports Hypotheses (H1)–(H3a–c). The results suggested that FP can increase firm CSR reporting by increasing GIP.

4.3.1. Robustness Tests with Fixed Effects

Hausman's test results determine whether the impact is fixed. Hausman tests have traditionally been used to test the consistency of OLS estimators when using pooled cross-section time series data. When applied to the data from the current study, this test yielded a significant result, demonstrating the utility of fixed-effect regression analysis. Consequently, before regression analysis, we ran Hausman's test on our model, and the fixed effect was applied to practically all models based on the findings of Hausman's test.

This robustness test was performed to check whether our significant finding that CSR reporting is associated with FP (ROE), GIP, and the ELS moderating this relationship is robust. FP (ROE) is calculated based on the division of net income by total assets, as in the previous study [77]. The authors further recommend that our average increase in firm GIP for our patient firms appears to occur after the reporting pledge becomes efficient. Taken together, the findings indicate that CSR reporting can encourage firm GIP.

The consequences of fixed effects and random effects in Models 1–5 are reported in Table 4. In Model 1, the value of FP (ROE) was positive and significant, and this coefficient of FP (ROE) has a positive impact on CSR reporting ($\beta = 3.054$, $p < 0.000$), which is in line with the literature [72,73], In Model 2, GIP is positively significant with CSR reporting association ($\beta = 0.046$, $p < 0.004$), which is in line with the literature [74,75]. Model 3 shows a negative coefficient, asserting that including the ELS as a moderator significantly increases and enhances CSR reporting (b = $-0.079$, $p < 0.000$), which is in line with the literature [76]. In Model 4, the estimation coefficients of the interaction term FP × ELS policy are all statistically negatively significant and have the same signs as those of CSR reporting ($\beta = -0.109$, $p < 0.001$), which indicates that the ELS enhances the relationship between CSR reporting. Such a finding is in line with the findings of [59]. Model 5 and Table 4 show that a significantly negative correlation exists between GIP × ELS and CSR reporting ($\beta = -1.105$, $p < 0.001$), Moreover, a significantly negative correlation can be observed between ELS and CSR reporting, which is in line with the findings of [59].

Overall, our robustness analysis indicates that GIP mediates this relationship, which is reliable with our key conclusions [76]. A robust examination of the relationship between GIP and FP and CSR reporting is provided in Table 5.

**Table 4.** Moderating effect of enterprise life stages on the relation between green innovation performance, firm performance, and CSR reporting (panel data analysis).

| | Fixed Effects | | | | |
|---|---|---|---|---|---|
| CSR Reporting | Model 1 FP (ROE) | Model 2 GIP | Model 3 ELS | Model 3 ELS | Model 5 GIP × ELS |
| FP (ROE) | 3.054 *** (0.643) | — | — | 9.246 *** (2.421) | — |
| GIP | — | 0.046 *** (0.066) | — | — | 0.085 *** (2.049) |
| ELS | | | −0.079 ** (0.047) | −0.037 (0.546) | −0.109 *** (0.682) |
| FP × ELS | — | — | — | −0.074 *** (0.058) | — |
| GIP ELS | | | | | −1.105 *** (8.620) |
| FA | 0.014 *** (2.874) | 0.014 *** (2.753) | 0.047 *** (2.784) | 1.580 ** (2.152) | 1.150 * (1.765) |
| FL | −1.13 *** (8.787) | −0.837 *** (6.277) | −0.783 *** (6.237) | 0.056 *** (2.784) | 0.017 *** (2.735) |

**Table 4.** *Cont.*

| CSR Reporting | Fixed Effects | | | | |
| --- | --- | --- | --- | --- | --- |
| | Model 1 FP (ROE) | Model 2 GIP | Model 3 ELS | Model 3 ELS | Model 5 GIP × ELS |
| SOEs | 0.814 ** | 0.918 ** | 0.857 ** | −0.724 *** | −0.856 *** |
| | (2.322) | (2.435) | (2.179) | (6.240) | (6.280) |
| CEOD | −0.374 | −0.413 | −0.437 | 0.860 ** | 0.906 ** |
| | (−0.948) | (−0.876 | (−0.874) | (2.172) | (2.440) |
| FS | 1.047 *** | 1.018 *** | 1.025 *** | −0.420 | −0.418 |
| | (41.896) | (40.788 | (40.877) | (−0.824) | (−0.854) |
| OC | 0.018 *** | 0.025 *** | 0.028 *** | 1.044 *** | 1.089 *** |
| | (3.235) | (0.296) | (0.342) | (40.823) | (40.744) |
| IO | −0.008 *** | −0.017 *** | −0.018 *** | 0.021 *** | 0.026 *** |
| | (3.977) | (3.787) | (3.876) | (0.380) | (0.284) |
| GO | 0.006 *** | 0.008 *** | 0.008 *** | −0.014 *** | −0.019 *** |
| | (0.763) | (7.379) | (7.239) | (−3.892) | (3.763) |
| YI | included | included | included | included | included |
| Constant | 10.336 *** | 09.979 *** | −7.649 *** | −11.645 *** | −11.960 *** |
| | (30.673) | (20.673) | (20.348) | (10.352) | (−13.686) |
| Hausman test Chi2 | 10.87 *** | 10.69 *** | 10.95 *** | — | — |
| | (0.047) | (0.076) | (0.062) | | |
| $R^2$ | 0.658 | 0.674 | 0.687 | 0.694 | 0.687 |

Note: *, **, ***, significant at 10%, 5%, and 1%, respectively.

### 4.3.2. Endogeneity Control

Our solution to the robustness problem utilized two alternative models. The researcher employed a one-year lagged model to re-measure corporate GIP and CSR reporting. Second, to address the robustness issue, a one-year-lagged measure of corporate GIP and CSR reporting is utilized in a two-stage least squares (2SLS) regression (a variable instrumental technique). The one-year-lagged and 2SLS regression results are shown in Table 5's 2SLS Models 1–6. Based on these results, it can be concluded that the results are robust.

After removing 2014 from our dataset, we restate our testing to run another test. As far as we can tell, if CSR reporting impacts FP (ROE) and GIP, removing the one-year sample should not influence the outcomes. These findings support the simultaneous pattern hypothesis. The results stated in Table 5 indicate that the interaction term significantly affects GIP.

The one-year lagged measure regression results are listed in Table 5 to test the moderating impact and thus reduce potential multicollinearity. Hypothesis (H3b,c) suggested that the ELS moderates the combined effect of FP (ROE) and

CSR reporting. Models 4–5 (Table 5) showed that the interaction term FP × ELS was negatively related to CSR reporting (β = −0.076, *p* < 0.001) and GIP × ELS (β = −0.075, *p* < 0.001), indicating that the ELS negatively moderates the effect of FP (ROA) and GIP on CSR reporting. Thus, Hypothesis (H3b,c) was supported [74,75].

The 2SLS regression results are listed in Table 5. Hypothesis (H3b,c) foretold that the ELS negatively moderates the relationship between the FP (ROA) qualities of CSR reporting. Models 9–10 (Table 5) showed that the interaction term FP × ELS was significantly negatively related to CSR reporting (β = −0.075, *p* < 0.001) and GIP × ELS (β = −1.107, *p* < 0.001), indicating that the ELS negatively moderates the effect of FP (ROE) and GIP on CSR reporting, thus supporting Hypothesis (H3b,c) [76].

**Table 5.** Moderating effect of enterprise life stages on the relation between green innovation performance, firm performance, and CSR reporting (robustness test one-year lagged and 2SLS).

| CSR Reporting | One-Year Lagged | | | | | 2SLS | | | | |
|---|---|---|---|---|---|---|---|---|---|---|
| | Model 1 FP (ROE) | Model 2 GIP | Model 3 ELS | Model 4 FP × ELS | Model 5 GIP × ELS | Model 6 FP (ROE) | Model 7 GIP | Model 8 ELS | Model 9 ELS | Model 10 GIP × ELS |
| FP (ROE) | 4.272 *** (4.853) | — | — | 9.438 *** (2.495) | — | 3.050 *** (0.642) | — | — | 9.257 *** (2.527) | — |
| GIP | — | 0.083 *** (2.417) | — | — | 0.085 *** (2.046) | — | 0.047 *** (0.075) | — | — | 0.087 *** (2.054) |
| ELS | — | — | −9.465 *** (2.452) | −0.036 *** (0.548) | −0.105 *** (0.684) | | | −0.085 ** (0.0537) | −0.038 (−0.573) | −0.112 *** (0.680) |
| FP × ELS | — | — | — | −0.076 *** (0.053) | — | — | — | — | −0.075 *** (0.059) | — |
| GIP × ELS | — | — | | — | −1.107 *** (8.625) | | | | | −1.107 *** (8.625) |
| FA | 1.435 * (1.844) | 1.150 * (1.765) | 1.582 ** (2.1475) | 1.584 ** (2.154) | 1.150 * (1.763) | 0.013 *** (2.875) | 0.017 *** (2.754) | 0.078 *** (2.786) | 1.582 ** (2.158) | 1.152 * (1.766) |
| FL | 0.014 *** (2.825) | 0.017 *** (2.732) | 0.057 *** (2.784) | 0.058 *** (2.783) | 0.017 *** (2.735) | −1.136 *** (8.785) | −0.838 *** (6.297) | −0.785 *** (6.238) | 0.057 *** (2.746) | 0.019 *** (2.734) |
| SOEs | −1.101 *** (8.733 | −0.855 *** (6.280) | −0.726 *** (6.249) | −0.724 *** (6.247) | −0.852 *** (6.280) | 0.817 ** (2.328) | 0.916 ** (2.437) | 0.856 ** (2.173) | −0.727 *** (6.244) | −0.857 *** (6.285) |
| CEOD | 0.857 ** (2.362) | 0.907 ** (2.443) | 0.867 ** (2.178) | 0.868 ** (2.177) | 0.907 ** (2.440) | −0.377 (0.945) | −0.417 (0.878) | −0.435 (0.876) | 0.862 ** (2.174) | 0.907 ** (2.452) |
| FS | −0.364 (0.970) | −0.418 (0.850) | −0.427 (−0.825) | −0.420 (0.824) | −0.418 (0.852) | 1.058 *** (41.872) | 1.019 *** (40.785 | 1.068 *** (40.823) | −0.405 (0.826) | −0.417 (0.858) |
| OC | 1.082 *** (41.883) | 1.087 *** (40.749) | 1.049 *** (40.802) | 1.048 *** (40.867) | 1.085 *** (40.746) | 0.019 *** (3.232) | 0.022 *** (0.297) | 0.025 *** (0.378) | 1.047 *** (40.884) | 1.051 *** (40.778) |
| IO | 0.017 *** (3.282) | 0.028 *** (0.286) | 0.027 *** (0.382) | 0.028 *** (0.387) | 0.029 *** (0.283) | −0.009 *** (3.978) | −0.019 *** (3.774) | −0.018 *** (3.862) | 0.020 *** (0.373) | 0.056 *** (0.286) |
| GO | −0.007 *** (3.943) | −0.019 *** (3.763) | −0.019 *** (3.893) | −0.013 *** (3.890) | −0.019 *** (3.763) | 0.008 *** (0.764) | 0.008 *** (7.372) | 0.008 *** (7.278) | −0.017 *** (3.890) | −0.018 *** (3.76) |
| YI | Included | included | included | included | included | included | included | included | included | included |
| Constant | 10.340 *** (14.683) | 11.960 *** (13.685) | −11.642 *** (10.354) | −11.649 *** (10.372) | −11.960 *** (13.672) | 10.378 *** (30.684) | 09.984 *** (20.635) | −7.656 *** (20.344) | −11.647 *** (10.394) | −11.974 *** (13.692) |
| $R^2$ | 0.682 | 0.689 | 0.697 | 0.695 | 0.689 | 0.655 | 0.678 | 0.683 | 0.697 | 0.692 |

Note: *, **, ***, significant at 10%, 5%, and 1%, respectively.

## 5. Contributions and Implications

This paper mainly aims to fill the gap between GIP and CSR reporting concerning business performance. It also investigates how the ELS may regulate CSR reporting initiatives and FP in this way. To achieve the aim of this paper, Chinese companies were selected. Over the last few decades, most companies have increased their efforts to sort out the issue of CSR reporting in the business world. This study's originality and importance stem from its varied contributions to the rapidly growing disciplines of corporate social governance (CSR) performance, financial analysis, and green innovation. First and foremost, this study provides a fresh viewpoint by investigating the moderating influence of a firm's life cycle on the complex interaction between firm performance, green innovation, and CSR reporting. This study adds to our understanding of how organizations negotiate CSR difficulties at various stages of growth by including the enterprise life stage as a contextual component. Furthermore, this study tackles the scarcity of empirical evidence in the literature by undertaking a rigorous analysis of green patent data from Chinese A-share-listed businesses over a long period of time, bridging the theoretical–practical divide. Furthermore, the findings of this study provide useful insights for both academics and practitioners, shining light on the unique intricacies of the interplay between financial and green innovation success, as well as their influence on CSR reporting. Finally, this research pushes the bounds of sustainability knowledge by providing a complete picture of how organizations could integrate financial, innovation, and sustainability objectives in pursuit of responsible and ethical business practices.

### 5.1. Theoretical Contributions

This study contributes theoretically to our knowledge of how the company life cycle and legitimacy theory meet with the complex dynamics of CSR reporting and performance. It emphasizes the need to take these contextual aspects into account while investigating the multidimensional linkages between firm performance, green innovation, and CSR reporting. This research provides significant theoretical contributions in two main areas: firm life cycle theory and legitimacy theory.

Firstly, this study adds to company life cycle theory by throwing light on the function of CSR reporting across the organization's life cycle. While previous research has focused largely on financial and strategic components of the company life stage, this study employs the enterprise life stage as a moderator to enhance the hypothesis. It highlights how a business's CSR performance and reporting are impacted by its developmental stage, offering a fresh look at the changing dynamics of CSR practices as a company grows and matures. This not only improves our understanding of the enterprise's life cycle dynamics but also highlights the complicated interplay between corporate sustainability activities and a company's overall evolutionary trajectory.

Second, within the context of legitimacy theory, this research contributes to our knowledge of how corporations strategically manage and coordinate their CSR reporting to retain social and environmental legitimacy throughout their life cycle. According to the legitimacy theory, organizations attempt to develop and maintain legitimacy in their social and environmental surroundings. This study experimentally verifies the legitimacy theory's applicability by finding that the breadth and character of a firm's CSR reporting varies dramatically throughout distinct life cycle stages. Companies may emphasize survival and acquiring money in the early phases of their life cycle, frequently focusing on financial success above CSR reporting. However, as businesses develop, they understand the importance of CSR reporting in maintaining their credibility and proactively include it in their entire business objectives. This insight adds to the legitimacy theory by demonstrating how corporations deliberately change their CSR reporting to retain their social and environmental legitimacy while negotiating the specific hurdles provided by various life cycle stages. These theoretical discoveries not only broaden our understanding of business sustainability but also lay the groundwork for future study in this area.

*5.2. Limitations*

First, this paper utilizes the CSMAR CSR rating as a metric to assess CSR reporting. Although robustness tests were conducted using Rankin's CSR rating, noticeable rating discrepancies persist among different rating agencies evaluating the same corporations. In future research, we aim to delve into the moderating impact of CSR rating disparities on CSR evaluation. Second, acknowledging the challenges in data collection, this study focuses on China's A-share-listed companies as research samples. It is important to note that empirical results might be influenced by selection bias. Therefore, expanding this study's scope to include SMEs is essential. This expansion will enable a deeper exploration of CSR awareness, performance, and the economic implications specific to SMEs.

Because China has various economic, cultural, and legislative aspects that might possibly affect CSR reporting in a divergent manner, the findings' generalizability to other locations or nations may be restricted. This study's temporal scope did not include the long-term effects of CSR efforts or the route of the evolution of corporate life phases, which might take a long time. This study may not have fully covered China's varied spectrum of industries, perhaps leading to divergent effects on business performance and green innovation in connection to CSR reporting across various sectors. The danger of biases being introduced in this study's sample selection approach stems from the probability that certain types of firms are more likely to report CSR data or go through various organizational life cycles. However, it should be emphasized that this study did not consider any changes in CSR reporting standards that may have occurred during the research period, thereby altering the results. This study excludes qualitative aspects of CSR reporting and green innovation success that have the potential to provide more meaningful insights.

*5.3. Future Research*

It is suggested that a comparative investigation of the moderating impact of enterprise life stages in other nations or areas be conducted to broaden the scope of this study. This will allow an assessment of the influence of different legislative and cultural settings on CSR reporting practice. It would be beneficial to use qualitative research approaches, such as conducting interviews or researching case studies, to gain a more in-depth understanding of the motives, difficulties, and solutions that firms use in the context of CSR reporting. Furthermore, investigating how these characteristics are impacted by the life cycle of businesses would add to a more thorough knowledge of the issue. It would recommend examining the effects of alterations in CSR reporting requirements and standards in China. We aim to investigate the moderating role of enterprise life stages on these consequences, as well as the techniques used by businesses to adjust to evolving reporting obligations. Future research might look at the moderating influence of enterprise life stages on stakeholder involvement and perception. This includes the impact on investment decisions, customer preferences, and business reputation. To improve the comprehensiveness of the study of the factors influencing CSR reporting, supplemental moderators, such as firm size, ownership structure, and management characteristics, must be included.

**Author Contributions:** Conceptualization F.R.; data curation, C.L.V.; formal analysis, F.R. and W.W.; methodology, F.R.; software, F.R.; writing—original draft, F.R.; writing—review and editing, W.W. and F.R. All authors have read and agreed to the published version of the manuscript.

**Funding:** This research received no external funding.

**Institutional Review Board Statement:** Not applicable.

**Informed Consent Statement:** Not applicable.

**Data Availability Statement:** The statistics supporting the outcomes of this research are accessible upon reasonable request from the first author.

**Conflicts of Interest:** The authors declare no conflicts of interest.

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
