# Peer review of "Interaction of Corporate Social Responsibility Reporting at the Crossroads of Green Innovation Performance and Firm Performance: The Moderating Role of the Enterprise Life Stage"

_sustainability, doi:10.3390/su16051821_

Round 1

Reviewer 1 Report

Comments and Suggestions for Authors

The topic of this article is of interest to wide readers. The paper's argument is built on an appropriate base of theory and the methods employed in the paper are appropriate. However, the abstract needs to contain enough information about the paper, and the author(s) should answer these questions about the manuscript: What was done? Why did you do it? What did you find? and Why are these findings useful and important?

The introduction should provide readers with the background information needed to understand your study and the reasons why you conducted your experiments. The introduction should answer the question: what question/problem was studied? the author(s) need to rewrite the introduction in this perspective.

The practical implications are not clearly identified in the text. The author(s) should address them in the revised version of the manuscript.

More serious work on the representative results must be undertaken. The way tables are represented in the text makes it less easy to understand the results. The findings in this paper are also briefly analyzed.

The author (s) should reference all the literature mentioned in the manuscript. The author (s) should ensure consistency in terms of results presentation, variable measurements, and models. 

Comments on the Quality of English Language

The standard of writing is not good enough for a journal like Sustainability. There are minor spelling and grammatical errors, and so the work would benefit from proofreading by a professional.

Author Response

The topic of this article is of interest to wide readers. The paper's argument is built on an appropriate base of theory and the methods employed in the paper are appropriate. However, the abstract needs to contain enough information about the paper, and the author(s) should answer these questions about the manuscript: What was done? Why did you do it? What did you find? and Why are these findings useful and important?

Answer: Thank you for your insightful feedback regarding the abstract of our paper. We appreciate your acknowledgment of the paper's relevance and the appropriateness of our theoretical base and methodology.

We understand the significance of an informative abstract and will ensure it adequately addresses the essential questions: What was done? Why did we pursue this research? What were the findings? And, importantly, why are these findings significant? To achieve this, we'll revise the abstract to provide a clear and comprehensive overview of our study. We'll explicitly outline the research objectives, methodology, key findings, and the broader implications of our results. Emphasizing the importance and usefulness of our findings within the context of the wider field will be a priority. Our aim is to ensure that the abstract serves as an effective summary, enabling readers to grasp the essence of the paper and understand its significance in addressing the research questions posed.

The introduction should provide readers with the background information needed to understand your study and the reasons why you conducted your experiments. The introduction should answer the question: what question/problem was studied? the author(s) need to rewrite the introduction in this perspective.

Answer: Thank you for your valuable insights regarding the introduction of our study. We appreciate your guidance on providing sufficient background information and context to help readers understand the motivations behind our research.

Understanding the importance of framing the study clearly, we'll revise the introduction to explicitly address the following questions: What question or problem did we investigate, and why did we conduct these experiments? Our aim is to present a comprehensive introduction that sets the stage by providing adequate background information, highlighting the specific problem or question addressed in our research, and elucidating the rationale behind conducting these experiments. By restructuring the introduction from this perspective, we aim to ensure that readers gain a clear understanding of the research problem, its significance, and the reasons driving our investigation.

The practical implications are not clearly identified in the text. The author(s) should address them in the revised version of the manuscript.

Answer: Thank you for highlighting the need to explicitly address the practical implications within our manuscript. We understand the importance of delineating these implications to bridge the gap between theoretical findings and real-world applications.

In our revised manuscript, we will ensure a dedicated section that explicitly outlines the practical implications derived from our study's findings. By doing so, we aim to illustrate how our research outcomes can be translated into actionable insights for practitioners, policymakers, or relevant stakeholders in the field. These implications will be carefully drawn from our findings, emphasizing their practical relevance, potential applications, and impact on decision-making processes in relevant domains. Your guidance is invaluable in ensuring that our research goes beyond theoretical significance and directly contributes to practical insights. Should you have any specific suggestions or areas where you believe practical implications should be emphasized.

More serious work on the representative results must be undertaken. The way tables are represented in the text makes it less easy to understand the results. The findings in this paper are also briefly analyzed.

Answer: Thank you for your valuable feedback regarding the representation and analysis of results in our paper. We acknowledge the importance of presenting findings in a clear, easily understandable manner and providing a comprehensive analysis.

To address this, we will undertake a more serious effort to improve the representation of tables in the text, aiming for greater clarity and ease of interpretation. Our goal is to present the results in a format that enhances readability and enables readers to grasp the findings more effectively. Additionally, we recognize the need for a more comprehensive analysis of the findings. In the revised manuscript, we will dedicate more attention to thoroughly analyzing the results, offering deeper insights, interpretations, and discussions to provide a robust understanding of the implications derived from the findings. Our commitment is to ensure that the results are presented in a more accessible format and that their analysis is elaborated upon in a manner that enhances the paper's overall quality and scholarly contribution.

The author (s) should reference all the literature mentioned in the manuscript. The author (s) should ensure consistency in terms of results presentation, variable measurements, and models. 

Answer: Thank you for your valuable feedback concerning referencing and ensuring consistency in our manuscript.

We will diligently review the manuscript to ensure that all mentioned literature is appropriately referenced and cited throughout the text. Our goal is to maintain consistency and accuracy in acknowledging the sources that contribute to our research. Additionally, we acknowledge the importance of consistency in presenting results, variable measurements, and models. We will thoroughly assess and revise the manuscript to ensure uniformity across these aspects, striving to present our findings and methodologies consistently and coherently. By maintaining consistency in these areas, we aim to enhance the overall quality and integrity of our manuscript, making it more accessible and credible for readers and reviewers. Should you have any further specific recommendations or areas where you believe consistency needs to be strengthened, please do not hesitate to share your insights. Your guidance is invaluable in refining our work.

Thank you once again for your valuable input.

Reviewer 2 Report

Comments and Suggestions for Authors

A: General Comments:

This study explores the impact of green innovation performance and firm performance on CSR reporting, with the Enterprise Life Stage (ELS) acting as a moderator. While the topic is relevant and timely, the study faces theoretical and methodological challenges. Here are specific comments for improvement:

B. Specific comments on each section

1- Abstract

Your abstract is informative but can be enhanced for clarity and completeness:

  • Specify the type of firm performance (e.g., financial, sustainability) when referencing it concerning CSR reporting.
  • Clearly define the term sustainability, which seems inconsistent throughout the paper.
  • Provide more details about the sample, including the industry breakdown.
  • Mention the use of 2SLS in addition to OLS, highlighting it as a robustness test

2- Introduction

Strengthen your introduction with the following suggestions:

  • Ensure seamless transitions between different concepts and sections.
  • Define acronyms like CSR consistently (Corporate Social Responsibility) and briefly explain terms like GIP and FP.
  • Explicitly state the research gap or problem your study addresses.
  • Preview key quantitative findings or trends to generate reader interest.
  • Clearly state research questions and objectives.
  • Offer more context on the sample's industry breakdown and why 2015-2021 was chosen.
  • Provide more detail on OLS regressions and panel data analysis, considering a more robust model like 2SLS or 3SLS.
  • Clarify why China was chosen as the focus of the study.
  • Emphasize implications for CSR reporting academics, practitioners, and regulators.
  • Ensure a smooth transition to the next section by summarizing main points.   3- Theoretical Framework and Hypothesis Development

Improve the theoretical framework with these suggestions:

  • Explicitly state the section's objectives.
  • Ensure smooth transitions between sub-sections.
  • Clearly define unfamiliar terms like ELS and GIP.
  • Take a clear stance on the theory used.
  • Integrate literature reviews seamlessly, citing specific studies.
  • Articulate logical linkages between CSR reporting, GIP, and firm performance.
  • Introduce quantitative terms where applicable.
  • Emphasize global relevance.   4- Data, Measurement, and Research Methodology

Enhance clarity in this section with the following:

  • Elaborate on why China was chosen and discuss generalizability.
  • Consider a more robust variable for firm performance, like Tobin’s Q.
  • Justify the sample selection and its relevance to the study.
  • Clarify the 2SLS stages and discuss the interaction with graphs.
  • Improve readability of the Correlation Matrix.
  • Provide more specifics on testing endogeneity, possibly redefining the model.   5- Conclusion

Complete the paper by addressing the conclusion section.

These comments aim to refine the paper and elevate its overall quality. Ensure coherence and clarity throughout, enabling readers to grasp the study's contributions effectively.

Comments on the Quality of English Language

The article is generally well-constructed, and the vocabulary used is appropriate for an academic context.

Author Response

(The authors gave the same response as above.)

Reviewer 3 Report

Comments and Suggestions for Authors

Interactive of CSR Reporting at the Crossroads of Green Innovation Performance and Firm Performance: Moderating Role of Enterprise Life Stage

I am pleased to see the revised version. The authors have done a very good job of improving the quality. It is a good and Well-written study and shows a new perspective on the topic under consideration. The topic of this research study has a practical significance to the scientific knowledge. The authors have investigated a good research area. I will accept this article after some changes. Modify according to these suggestions.

Before accepting this study for publication, I suggest changes to improve the quality. I need strong literature support to reach merit for publication. I suggest the authors cite these studies to improve the quality. See the below recommended studies to improve your abstract quality. I suggest the studies below to improve the quality. I will accept this paper after these changes. 

Decision: Improve the quality and submit a revised paper. I will accept it for publication after changes.

Title

The title can be improved.

First, I have some suggestions for the authors to enhance the quality of this innovative study. Please write a high-quality abstract, as it is the main door of the study. I suggest authors remove some type errors and make it in a meaningful way to reflect the whole idea. Remove minor grammar errors.

The introduction section needs improvement. Please read these studies, revise your abstract, and cite them in the introduction and literature part. Cite the suggested studies to improve the quality. The introduction is not well established with the support of the study objectives and fresh literature evidence. The introduction should benefit to execute further improvement in the organization and clarity of the study argument. I invite the author to define the topic's gap and indicate how the paper fills the gap. Cite these studies to strengthen the quality of this study.

I suggest removing some typo errors from the literature. This review of the study needs improvement with the latest support of the literature. Thus, it is necessary to update the literature review part with fresh studies. At the end of the introduction, please indicate a theoretical contribution of the paper and add the structure of the paper by sections. I recommend the author find the literature gap and indicate how the paper fills that gap. Cite these studies to strengthen the quality of this study.

The methods section could be more explicit about the specific research question or hypotheses being tested. Furthermore, the methodology could be more detailed about the statistical methods used to analyze the data. Additionally, the section could benefit from more information about how missing data was handled, what assumptions were made about the distribution of the outcome variables, and how the model fit was assessed. I recommend including the data part inside the methodology and naming the section Data and Methodology. Cite more latest studies in the methodology to improve the study.

The method section could benefit from further improvement. It is important to provide a clear justification for the methodology approach used, explaining why it was chosen and how it is appropriate for the research question at hand. Additionally, it would be helpful to reference prior studies that have successfully used this methodology approach to strengthen the argument for its use in this particular study.

The data section requires improvement. The authors must address several key questions to provide a better understanding of their approach. Specifically, why were these variables selected for the model? What does the existing literature say about these variables? Besides, it is vital to provide information on previous authors who have used these variables. Without this information, readers may find it difficult to fully comprehend the approach and results presented in the study.

I recommend the author add robustness test results to verify the results' correctness. Besides, it is necessary to add a discussion part to interpret the result obtained in detail.

Explain this section effectively.  It needs a better presentation related to the study topic. Discuss the study’s limitations with a separate heading and discuss it briefly. Policy recommendations are not sufficient at this stage of the manuscript. The authors must add a separate section for policy recommendations in the conclusion section. Also, add some exciting limitations regarding political factors for future studies. 

The conclusion provides a comprehensive summary of the findings, the conclusion could have been strengthened by acknowledging some limitations of the study, such as potential confounding variables that may have affected the results. The conclusion could have also provided some recommendations for future research. I recommend the author compare the results obtained in the study with the findings of other authors and explain how the results relate to each other.

Comments on the Quality of English Language

Moderate changes

Author Response

Comments and Suggestions for Authors

Interactive of CSR Reporting at the Crossroads of Green Innovation Performance and Firm Performance: Moderating Role of Enterprise Life Stage

I am pleased to see the revised version. The authors have done a very good job of improving the quality. It is a good and Well-written study and shows a new perspective on the topic under consideration. The topic of this research study has a practical significance to the scientific knowledge. The authors have investigated a good research area. I will accept this article after some changes. Modify according to these suggestions.

Before accepting this study for publication, I suggest changes to improve the quality. I need strong literature support to reach merit for publication. I suggest the authors cite these studies to improve the quality. See the below recommended studies to improve your abstract quality. I suggest the studies below to improve the quality. I will accept this paper after these changes. 

Decision: Improve the quality and submit a revised paper. I will accept it for publication after changes.

Title

The title can be improved.

First, I have some suggestions for the authors to enhance the quality of this innovative study. Please write a high-quality abstract, as it is the main door of the study. I suggest authors remove some type errors and make it in a meaningful way to reflect the whole idea. Remove minor grammar errors.

Answer: Thank you for your thoughtful suggestions on enhancing the quality of our study’s abstract. We appreciate your input and recognize the importance of a strong abstract as the gateway to our research. We will diligently revise the abstract to ensure it accurately encapsulates the essence of our study, focusing on removing typographical errors and refining the language to convey our ideas more meaningfully. Addressing minor grammatical errors is also a priority to present a polished and professional abstract.

Answer: The introduction section needs improvement. Please read these studies, revise your abstract, and cite them in the introduction and literature part. Cite the suggested studies to improve the quality. The introduction is not well established with the support of the study objectives and fresh literature evidence. The introduction should benefit to execute further improvement in the organization and clarity of the study argument. I invite the author to define the topic's gap and indicate how the paper fills the gap. Cite these studies to strengthen the quality of this study. We will thoroughly review the suggested studies to bolster our understanding and revise the abstract to incorporate citations that strengthen the introduction and literature review. It's our priority to establish a clear linkage between the study objectives and the existing literature, addressing any gaps in the field. Additionally, we acknowledge the importance of defining the topic's gap and illustrating how our paper contributes to filling that void. We will ensure that the introduction effectively delineates this gap and articulates how our study bridges it, providing clarity and organization to the overall argument.

I suggest removing some typo errors from the literature. This review of the study needs improvement with the latest support of the literature. Thus, it is necessary to update the literature review part with fresh studies. At the end of the introduction, please indicate a theoretical contribution of the paper and add the structure of the paper by sections. I recommend the author find the literature gap and indicate how the paper fills that gap. Cite these studies to strengthen the quality of this study.

Answer: Thank you for your meticulous review and valuable suggestions regarding the literature section of our study. We appreciate your insights and understand the importance of ensuring accuracy and relevance in citing the latest studies. We will diligently address any typographical errors present in the literature review and update it with the most recent and pertinent studies available. It's crucial for us to ensure that the review reflects the current landscape of research in our field. Regarding the conclusion of the introduction, we recognize the significance of outlining the theoretical contribution of our paper and providing a clear structure for the subsequent sections. We will incorporate a concise statement highlighting the paper's theoretical contribution and structure, enhancing the overall coherence of the study. Furthermore, your suggestion to identify the literature gap and illustrate how our paper fills that void is invaluable. We'll ensure the introduction succinctly identifies this gap and delineates how our study contributes to addressing it, substantiating our work within the field.

The methods section could be more explicit about the specific research question or hypotheses being tested. Furthermore, the methodology could be more detailed about the statistical methods used to analyze the data. Additionally, the section could benefit from more information about how missing data was handled, what assumptions were made about the distribution of the outcome variables, and how the model fit was assessed. I recommend including the data part inside the methodology and naming the section Data and Methodology. Cite more latest studies in the methodology to improve the study.

Answer: Thank you for your comprehensive feedback on the methods section of our study. Your suggestions to enhance the explicit nature of our research questions or hypotheses, provide more detailed insight into statistical methods, and address aspects related to missing data handling and model assessment are greatly appreciated. We acknowledge the importance of explicitly outlining the research questions or hypotheses being tested. We will revise this section to ensure clarity and precision in articulating our research focus. Additionally, we'll provide more comprehensive details about the statistical methods employed for data analysis, including information about missing data handling, assumptions regarding outcome variable distribution, and a thorough assessment of model fit. Your recommendation to merge the data details within the methodology and rename the section as Data and Methodology aligns with our goal of providing a more cohesive and comprehensive description of our research process. This adjustment will undoubtedly enhance the clarity and organization of our methodology section. Furthermore, your suggestion to cite more recent studies within the methodology is invaluable. We will incorporate these references to strengthen the methodological framework and ensure our study is firmly situated within the current scholarly discourse.

The method section could benefit from further improvement. It is important to provide a clear justification for the methodology approach used, explaining why it was chosen and how it is appropriate for the research question at hand. Additionally, it would be helpful to reference prior studies that have successfully used this methodology approach to strengthen the argument for its use in this particular study.

Answer: Thank you for your insightful feedback regarding the method section of our study. Your emphasis on the importance of providing a clear rationale for our chosen methodology and referencing prior studies to support its appropriateness is invaluable. We understand the significance of explicitly justifying our selected methodology, outlining why it was chosen, and how it aligns with the research question at hand. We will revise this section to offer a comprehensive rationale, highlighting the suitability of our chosen approach for our specific research objectives. Furthermore, we recognize the importance of referencing prior studies that have effectively utilized a similar methodology. Incorporating these references will not only strengthen our argument for employing this methodology but also highlight its successful application in similar research contexts.

The data section requires improvement. The authors must address several key questions to provide a better understanding of their approach. Specifically, why were these variables selected for the model? What does the existing literature say about these variables? Besides, it is vital to provide information on previous authors who have used these variables. Without this information, readers may find it difficult to fully comprehend the approach and results presented in the study.

I recommend the author add robustness test results to verify the results' correctness. Besides, it is necessary to add a discussion part to interpret the result obtained in detail.

Explain this section effectively.  It needs a better presentation related to the study topic. Discuss the study’s limitations with a separate heading and discuss it briefly. Policy recommendations are not sufficient at this stage of the manuscript. The authors must add a separate section for policy recommendations in the conclusion section. Also, add some exciting limitations regarding political factors for future studies. 

Answer: Thank you for your thorough review and constructive feedback on the data section of our study. Your suggestions to address key questions regarding variable selection, referencing existing literature, and providing information about previous authors using these variables are invaluable. We acknowledge the need to elucidate why specific variables were chosen for our model and to ground this selection in the existing literature. We will enhance this section by providing a detailed rationale for variable selection, citing relevant literature to support our choices, and referencing previous studies that have successfully utilized these variables. This addition will offer readers a deeper understanding of our approach and results. Regarding robustness testing, we appreciate your suggestion and recognize its significance in verifying the correctness of our results. We will incorporate robustness tests to strengthen the credibility of our findings and ensure their reliability. Furthermore, your recommendation to include a dedicated discussion section to interpret the results in detail aligns with our aim to provide a comprehensive understanding of our findings. This section will effectively present the results in relation to the study topic and contribute to a more nuanced interpretation. Addressing limitations is crucial, and we will include a separate heading discussing the study's limitations, providing a concise yet thorough overview. Additionally, we understand the importance of a distinct section for policy recommendations in the conclusion. We'll restructure the conclusion to include a dedicated segment for policy recommendations, ensuring a more organized presentation. Lastly, your suggestion to highlight intriguing limitations related to political factors for future studies is appreciated. We will incorporate these intriguing limitations for potential future research avenues.

The conclusion provides a comprehensive summary of the findings, the conclusion could have been strengthened by acknowledging some limitations of the study, such as potential confounding variables that may have affected the results. The conclusion could have also provided some recommendations for future research. I recommend the author compare the results obtained in the study with the findings of other authors and explain how the results relate to each other.

Answer: Thank you for your insightful feedback on the conclusion section of our study. Your recommendations to further strengthen the conclusion by acknowledging limitations, providing recommendations for future research, and comparing our results with those of other authors are valuable insights. We recognize the importance of acknowledging potential limitations, such as confounding variables, that might have impacted our results. We will revise the conclusion to include a discussion on these limitations, offering a more comprehensive view of the study's scope and potential areas for further investigation. Additionally, your suggestion to incorporate recommendations for future research aligns with our goal of contributing to the advancement of knowledge in this field. We'll ensure the conclusion includes specific suggestions for future research directions, highlighting areas that could benefit from additional investigation based on our study's outcomes. Furthermore, comparing our findings with those of other authors is crucial for contextualizing our results within the existing body of literature. We'll revise the conclusion to include a comparative analysis, elucidating how our results relate to and complement the findings of other studies. Your guidance will undoubtedly enhance the depth and relevance of our conclusion section. Should you have additional recommendations or specific studies you believe would complement our comparative analysis, we would greatly appreciate your continued support.

Thank you once again for your valuable feedback.
